# X-Streamer: Unified Human World Modeling with Audiovisual Interaction

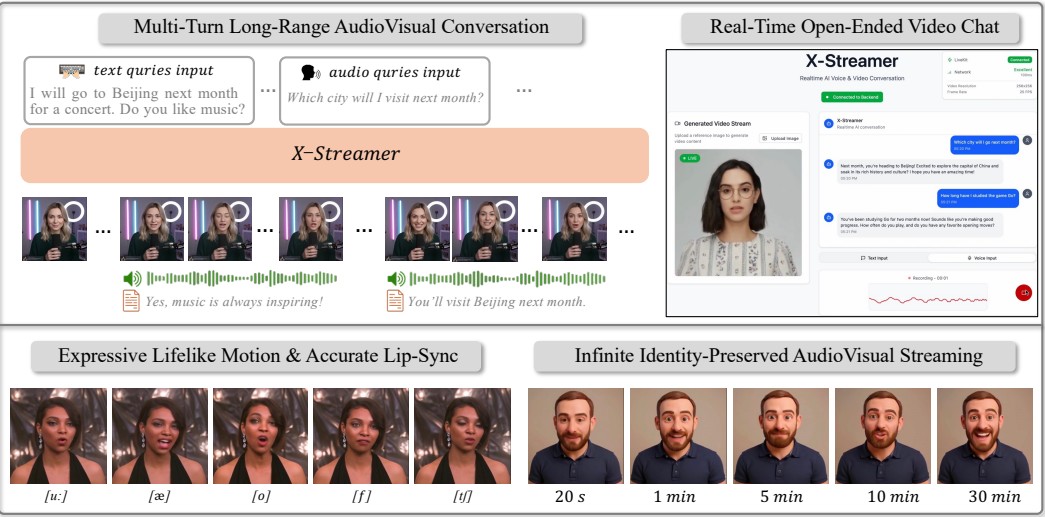

Figure 1: We present X-Streamer, a framework that constructs an infinitely streamable digital human from a single portrait, capable of generating intelligent, real-time, multi-turn responses across text, speech, and video. X-Streamer delivers phoneme-level lip synchronization while maintaining long-range conversational memory and visual consistency throughout extended audiovisual interactions.

## Abstract

We introduce X-Streamer, an end-to-end multimodal human world modeling framework for building digital human agents capable of infinite interactions across text, speech, and video within a single unified architecture. Starting from a single portrait, X-Streamer enables real-time, open-ended video calls driven by streaming multimodal inputs. At its core is a Thinker–Actor dual-transformer architecture that unifies multimodal understanding and generation, turning a static portrait into persistent and intelligent audiovisual interactions. The Thinker module perceives and reasons over streaming user inputs, while its hidden states are translated by the Actor into synchronized multimodal streams in real time. Concretely, the Thinker leverages a pretrained large language–speech model, while the Actor employs a chunk-wise autoregressive diffusion model that cross-attends to the Thinker's hidden states to produce time-aligned multimodal responses with interleaved discrete text and audio tokens and continuous video latents. To ensure long-horizon stability, we design inter- and intra-chunk attentions with time-aligned multimodal positional embeddings for fine-grained cross-modality alignment and context retention, further reinforced by chunk-wise diffusion forcing and global identity referencing. X-Streamer runs in real time on two A100 GPUs, sustaining hours-long consistent video chat experiences from arbitrary portraits and paving the way toward unified world modeling of interactive digital humans.

## 1 Introduction

Recent advancements in generative AI have enabled the creation of coherent conversational text and speech Schulman et al. (2022); Comanici et al. (2025); Zeng et al. (2024); Grattafiori et al.

(2024), as well as aesthetically pleasing images and videos Esser et al. (2024); Hurst et al. (2024); Google DeepMind (2025); Wan et al. (2025); Kuaishou Technology (2025); Kong et al. (2024); Gao et al. (2025) from diverse conditional prompts such as text, speech, and camera poses. In parallel, world models Bruce et al. (2024); Ball et al. (2025); Assran et al. (2025); Agarwal et al. (2025); Team et al. (2025a); Song et al. (2025a) have emerged as a fundamental paradigm for understanding and generating complex environments, supporting long-range interactive explorations. However, for digital human agents, we envision a new generative paradigm with two key capabilities: (1) infinite streaming multimodal interaction while retaining long-range multi-turn context, and (2) first-person self-evolvement and audiovisual engagement with intelligent, context-aware responses. Building such human agents has transformative potential across entertainment, live streaming, education, shopping and agents, yet achieving this level of open-ended, cross-modal interaction remains a formidable challenge. In this work, we introduce a novel generative paradigm for human agent world modeling, with a focus on conversational audiovisual interactions at the head-portrait scale.

Existing systems for interactive human agents are often built on sequential, modular pipelines, where separate models handle conversational text and speech generation, as well as video animation. While such modular designs enable specialization within each modality, they come with inherent drawbacks: unidirectional contextual flow, latency in multimodal generation, and reliance on handcrafted control logic for temporal and semantic alignment across modalities. These limitations become especially pronounced in long-form audiovisual interactions, where maintaining consistency in identity, motion, and context is challenged by compute and memory constraints, along with error accumulation over time. In contrast, unified understanding and generation frameworks have shown strong in-context learning, multitask generalization and tighter cross-modality alignment. However, prior work has largely concentrated on text-speech Xu et al. (2025); AI et al. (2025); Huang et al. (2025a); Zeng et al. (2024) and text-image generation Deng et al. (2025); Wu et al. (2025); Wang et al. (2024); Ge et al. (2024); Team et al. (2025b); Chen et al. (2025b), leaving the space of omnimodal understanding and generation, spanning text, speech and video, largely unexplored.

In this work, we propose X-Streamer, a multimodal human world modeling framework that jointly understands and generates text, speech, and video within a single unified architecture, trained end-to-end on unlabeled human talking videos. Given a single portrait image and streaming user queries in text or audio form, the model generates synchronized and context-aware text, speech, and video responses in real time, enabling extended multi-round audiovisual conversations. The core challenges are threefold: (1) unifying and synchronizing multimodal streaming generation across continuous video tokens and discrete text and audio tokens, (2) maintaining persistent audiovisual consistency over long-range context, and (3) ensuring real-time efficiency for interactive multimodal generation.

To achieve this, we adopt a Thinker–Actor architecture, inspired by Qwen2.5-Omni Xu et al. (2025), which mirrors human cognition and behavior through synergistic dual-track multimodal autoregressive models. The Thinker module leverages a pretrained language–speech model Zeng et al. (2024) to provide conversational intelligence by interpreting user intent from streaming text and audio queries. Its hidden embeddings are then autoregressively translated by the Actor, a learnable module also initialized from a pretrained language model Zeng et al. (2024), into interleaved discrete text and audio tokens alongside continuous video latent tokens. Our design preserves the pretrained language–speech capabilities while extending them to the video modality through autoregressive diffusion in a continuous latent space. To satisfy real-time constraints while maintaining long-range temporal coherence, we adopt a highly compressed video VAE latent tokenization HaCohen et al. (2024). Within the Actor, temporal continuity and semantic alignment across modalities are enforced by cross-attention between the Thinker's audio–text hidden states and the visual tokens. All outputs are temporally synchronized using a unified 3D multimodal rotary positional embedding (RoPE) and generated in an interleaved manner to minimize latency. For long-horizon stability, we employ a chunk-wise diffusion-forcing scheme Chen et al. (2024) and an optimized inference-time noise scheduler, reinforced by lightweight global reference image conditioning.

Our model comprises 18B parameters and is trained on $4,248.6$ hours of talking-head videos. With inference-time optimizations, we show that our approach supports real-time, open-ended multimodal interaction on two A100 GPUs, producing infinite audiovisual streams that preserve long-range conversational coherence, characteristic identity, and expressive alignment across speech and motion. This work marks a step toward lifelike, persistent, and intelligent human agents capable of seamless engagement in complex multi-turn conversations.

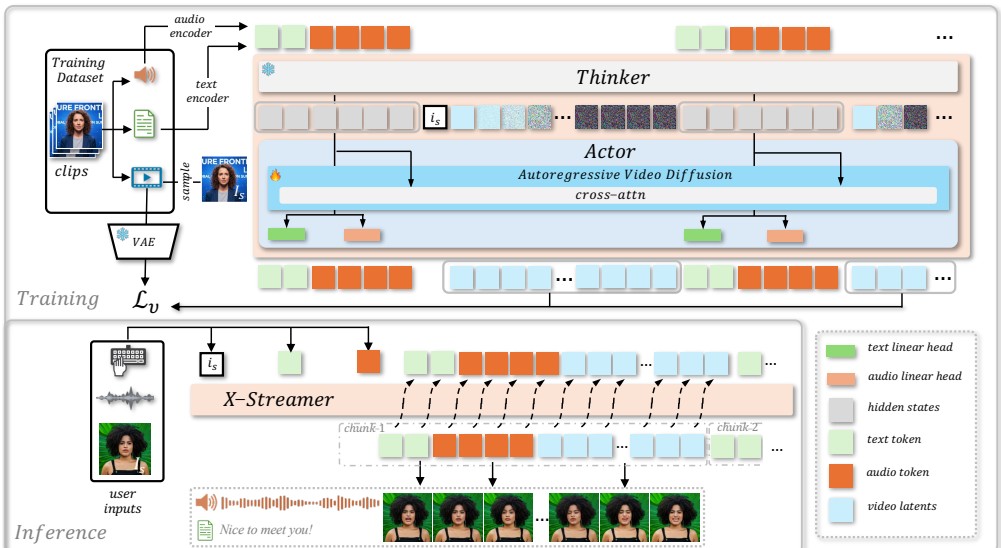

Figure 2: **Overview of X-Streamer**. Given a single portrait $I_s$, X-Streamer enables real-time audio-visual interaction through a dual-track autoregressive framework. A frozen Thinker transformer, instantiated from a pretrained language–speech model, interprets streaming user text and audio queries, while an Actor generates synchronized interleaving text, speech, and video streams from the Thinker's hidden states. Video is produced with chunk-wise autoregressive diffusion stabilized by diffusion forcing, and multimodal alignment is enforced via cross-attention. Deployed on two A100 GPUs, X-Streamer streams at 25 fps, enabling coherent, long-horizon multimodal interactions.

## 2 RELATED WORK

**Autoregressive Video Diffusion.** Diffusion models Ho et al. (2020); Song et al. (2020); Rombach et al. (2022) have become the dominant paradigm for video generation, training models to iteratively denoise sequences from noisy inputs. Existing approaches Ho et al. (2022); Blattmann et al. (2023a); Mei & Patel (2023); Ma et al. (2024); Yang et al. (2024); Wan et al. (2025); Kong et al. (2024); Gao et al. (2025) typically adopt uniform-step schedulers during training and inference to preserve temporal consistency, but their reliance on fixed-length sequences limits scalability to streaming settings with variable horizons. Chunk-wise diffusion models Blattmann et al. (2023b); Chen et al. (2023); Luo et al. (2023); Voleti et al. (2022) extend sequence length via sliding windows, yet still suffer motion and semantic discontinuities due to restricted context. Autoregressive approaches Yan et al. (2021); Hong et al. (2022); Ge et al. (2022); Yu et al. (2023); Kondratyuk et al. (2023) instead generate frames sequentially conditioned on past outputs, but error accumulation under teacher forcing Rasul et al. (2021) leads to drift and quality degradation over long horizons. Recent work mitigates this mismatch through self-forcing strategies Huang et al. (2025b); Lin et al. (2025b), narrowing the training–inference gap. Asynchronous diffusion methods Chen et al. (2024); Song et al. (2025b); Liu et al. (2024b); Sun et al. (2025); Kodaira et al. (2025); Teng et al. (2025); Chen et al. (2025a) further enhance robustness by applying independent noise schedules per frame, reducing drift and corruption across extended sequences. Building on these advances, we unify multimodal generation with chunk-wise autoregressive video diffusion under asynchronous noise Chen et al. (2024), enabling infinite-horizon, real-time multimodal interaction for digital humans.

**Real-Time AudioVisual Interaction.** Recent language–speech models Zeng et al. (2024); Du et al. (2024); AI et al. (2025); Team (2024); Xu et al. (2025) have achieved low-latency, context-aware spoken interactions. Extending these capabilities to audiovisual responses in real time, however, remains challenging. Most existing methods Zhu et al. (2025); Low & Wang (2025); Xu et al. (2024b); Chen et al. (2025c) adopt modular pipelines, where a speech generation model is paired with a talking-head renderer to produce audio-driven videos. Recent advances in portrait animation have improved expressiveness, either through intermediate facial motion representations Zhang et al. (2023); Wang et al. (2023); He et al. (2023); Ma et al. (2023); Zhang et al. (2025b); Xu et al. (2024b); Zhang et al. (2025a) or via end-to-end training Tian et al. (2024); Jiang et al. (2024); Xu et al. (2024a); Wang et al. (2025a); Lin et al. (2025a). While these methods achieve lip synchronization, they rely exclusively on acoustic cues and lack multi-turn conversational memory and semantic

reasoning. To mimic dyadic conversations, some works have also explored generating "listening states" Zhou et al. (2022); Liu et al. (2024a); Zhou et al. (2025); Tran et al. (2024). More recently, Veo3 Google DeepMind (2025) and OmniTalker Wang et al. (2025b) generate speech and video jointly, yet still depend on externally provided text inputs for content. In contrast, our approach unifies multimodal understanding and generation in a single framework, enabling digital humans that can listen, think, and act—producing context-aware audiovisual responses in real time Ao (2024).

## 3 METHOD

In this work, we aim to build a lifelike human agent that can listen, speak, and act, starting from a single portrait image $I_s$. Given streaming, multi-turn user queries in the form of text $T_i$, audio $A_i$, or their combination, the agent generates coherent, context-aware responses with synchronized text $T_o$, audio $A_o$, and video $V_o$. We frame this task as a generative world modeling paradigm for digital humans, characterized by its capability to support infinite audiovisual generation with long-range context, self-adaptive evolvement and real-time user interaction.

In Section 3.1, we first introduce a unified world modeling formulation based on synergistic dual transformers, where an Thinker transformer performs understanding and reasoning over user queries $(T_i, A_i)$, while an Actor transformer translates the hidden states of the Thinker into interleaved, time-aligned responses $(T_o, A_o, V_o)$. Our design largely inherits pretrained language–speech understanding and generation capabilities, while we provide details on extending to streaming video generation in Section 3.2. To ensure long-range coherent visual generation, we integrate a chunk-wise diffusion-forcing scheme and reference context management into the autoregressive video diffusion process, which is further optimized to support real-time multimodal inference on two A100 GPUs.

### 3.1 UNIFIED HUMAN WORLD MODELING

We formulate the task of building interactive digital human agents as a unified multimodal understanding and generation problem, defined as

$$(T_o, A_o, V_o) = \mathcal{M}(T_i, A_i, I_s), \tag{1}$$

where $\mathcal{M}$ denotes a transformer-based multimodal autoregressive model. Here, $I_s$ is the static portrait image depicting the agent's appearance, $(T_i, A_i)$ are streaming user queries in text and audio form, and $(T_o, A_o, V_o)$ are the corresponding multimodal responses of text, audio, and video. The model is trained autoregressively over a unified token sequence that interleaves text, audio and video. For text and audio, we employ pretrained tokenizers and decoders following Zeng et al. (2024), where both modalities are encoded into discrete semantic tokens, denoted as $t$ and $a$ respectively. For video, we adopt the compact LTX HaCohen et al. (2024) VAE latent code $v$ with $8 \times 32 \times 32$ spatiotemporal compression ratio, facilitating real-time video generation with long-horizon context. The training objective is then to maximize the likelihood of the target multimodal response at time $c$, conditioned on the given reference image, user queries, and the generated multimodal history:

$$\mathcal{L} = -\log P(t_o^c, a_o^c, v_o^c \mid i_s, t_i^{<c}, a_i^{<c}, t_o^{<c}, a_o^{<c}, v_o^{<c}), \tag{2}$$

where the superscript $< c$ denotes preceding tokens across modalities, and $i_s$ is the encoded latent of the reference image $I_s$ from the visual encoder.

**Thinker-Actor Dual-Transformer Architecture** Training such a multimodal transformer from scratch would require an enormous corpus of data that spans all modalities for pretraining, along with multi-turn conversational speech–video pairs for instruction finetuning. Both are extremely difficult to curate at scale. Achieving high-quality generation across text, speech, and video also demands a delicate balance of heterogeneous datasets, such as text–speech, speech–video, and text–speech–video. In contrast, many pretrained LLMs and LSMs already possess strong multi-turn conversational text-speech capabilities. By leveraging these pretrained models, we inherit their reasoning and conversational intelligence while extending them into the video modality, enabling a unified framework for multimodal understanding and generation.

To achieve this, we draw inspiration from the human cognitive process of interpreting information, formulating responses and executing actions. Accordingly, we design $\mathcal{M}$ as a dual-transformer architecture (Figure 2) consisting of a Thinker and an Actor, similar to the paradigm of Qwen2.5-Omni Xu et al. (2025). The Thinker is instantiated with GLM-4-Voice Zeng et al. (2024) and kept frozen, preserving its pretrained conversational intelligence across text and speech. The Actor, composed of modality-specific generators, consumes the streaming hidden states produced by

the Thinker and translates them into synchronized multimodal outputs chunk by chunk, operating on two-second segments of text, audio, and video. Specifically for text and speech, a linear head projects the hidden states into discrete tokens, which are further processed by a conditional flow-matching model Lipman et al. (2023) and a HiFi-GAN vocoder Kong et al. (2020) to synthesize speech waveforms. For video, we train a parallel transformer, also initialized from the weights of GLM-4-Voice, to autoregressively predict video token sequences given the Thinker's hidden states. Notably, this video transformer can differ architecturally from the Thinker, while initialization with pretrained LLM weights significantly improves convergence and training stability.

**Time-Aligned MultiModal Generations** We follow the streaming generation paradigm of GLM-4-Voice, where the transformer alternates between 13 text tokens and 26 speech tokens, corresponding to a roughly 2-second window given the 12.5 Hz speech tokenizer. We extend this scheme to three modalities by introducing video tokens into the sequence. Specifically, after every 26 speech tokens, the Actor generates $(\frac{26}{12.5} \times 25)/8 \times \frac{H}{32} \times \frac{W}{32}$ video tokens, representing a 25-fps 2.08-second video segment at resolution $H \times W$. This chunk-wise interleaving allows video to be generated under the full guidance of text–audio semantics, achieving tight audio–visual temporal alignment. At the same time, it minimizes video generation latency by eliminating the need to buffer the entire speech output, as required in modular audio-driven video generation approaches.

Audio–visual synchronization, manifested through accurate lip-sync and speech-expression alignment, is essential for building lifelike interactive humans. To this end, we introduce two key designs in the video generation transformer within the Actor. First, rather than using a shared self-attention across modalities as in Mixture of Transformer Liang et al. (2024); Shi et al. (2024), we incorporate a cross-modal attention layer after each self-attention layer in every transformer block, conditioning the video tokens prediction on the corresponding chunk of text–audio hidden states. Second, in addition to applying 3D RoPE Su et al. (2024); Heo et al. (2024) to video tokens indicating spatiotemporal position, we assign 1D RoPEs to the conditional text-audio hidden embeddings, aligned along the temporal axis. Together, these strategies enforce explicit chunk-wise temporal correspondence between audio and video, leading to improved lip-sync and more natural audio–visual alignment.

**Audio-Visual Context Attentions.** For text and speech, we leverage GLM-4-Voice's pretrained ability to maintain up to 8K tokens of multi-turn conversational context, where the Actor's generated text and audio tokens are routed back into the Thinker. Since GLM-4-Voice does not accept visual inputs, video tokens are handled solely within the Actor. Visual context is preserved via self-attention in the video transformer, ensuring semantic consistency and pixel continuity, while conversational context is injected through cross-attention with the Thinker's hidden states. With the $8\times$ temporal compression of our video VAE, we treat 8 frames as the basic generation unit. Within each unit, we apply bidirectional self-attention and full cross-attention to the aligned text–audio states. Across units, causal attention over preceding video tokens enforces temporal causality, enabling coherent chunk-by-chunk video stream generation. For clarity, we refer to each 8-frame unit as a video chunk, with all modalities interleaved and generated within roughly 2-second windows.

## 3.2 REAL-TIME STREAMING VIDEO GENERATION

Our dual-track transformers (Section 3.1) enable conversational context-aware video generation, yet three challenges remain. First, unlike discrete text and speech tokens trained with cross-entropy under teacher forcing, video is represented as continuous latent embeddings that are less native to autoregressive generation. Second, long-range video generation is vulnerable to error accumulation, often causing drift or corrupted frames after only a few chunks. Third, despite using highly compressed VAE HaCohen et al. (2024) at medium resolution, video still requires far more tokens than text and speech, posing significant challenges for low-latency, omni-modal generation.

**Chunk-Wise Autoregressive Video Diffusion** To unify continuous video latent generation within the autoregressive framework alongside text and speech, we employ a diffusion-based objective. At each step, the Actor predicts the next chunk of video latents through iterative denoising, conditioned on previously generated video embeddings. Formally, let $v^c$ denote the $c$-th chunk of video latent embeddings, and $v_k^c$ its noisy counterpart obtained by corrupting $v^c$ with Gaussian noise over $k$ diffusion steps. We adopt the velocity prediction (v-prediction) parameterization, where the model is trained to predict the target velocity vector $vel_k^c$. The training loss is given by:

$$\mathcal{L}_v = \mathbb{E}_{v^c, k, \epsilon \sim \mathcal{N}(0,I)} \left[ \left\| \hat{vel}^c - vel_k^c \right\|^2 \right], \qquad \hat{vel} = vel_\theta(v_k^c(\epsilon), k, h^c, v^{<c}), \qquad (3)$$

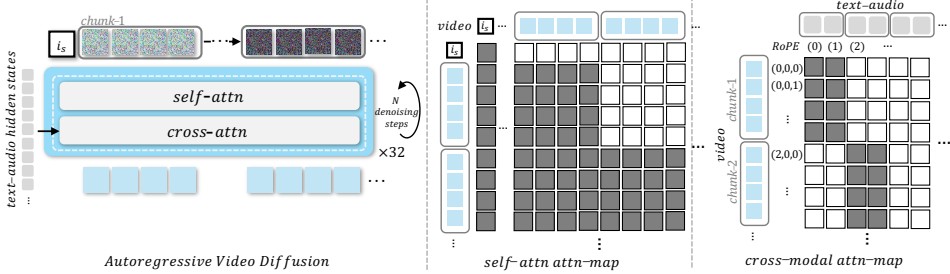

Figure 3: **Autoregressive Video Diffusion.** The video transformer generates video chunk by chunk, applying bidirectional spatial self-attention within each chunk and cross-attention to the Thinker's text–audio hidden states, while enforcing causal temporal attention across chunks. Global attention to the reference image is maintained throughout. To stabilize long-horizon generation, we adopt chunk-wise diffusion forcing by assigning independent noise levels across chunks.

where $vel_\theta$ is the model's predicted velocity given the noisy latents, the diffusion timestep $k$, the corresponding Thinker's hidden states $h^c$, and video latent history $v^{<c}$. During inference, we follow the DDIM scheduler Song et al. (2020) to iteratively denoise the video chunk from Gaussian noise.

For our video generation backbone, we adopt the GLM-4-Voice architecture as the Thinker and initialize training from its pretrained weights. To integrate video latents into the language-model backbone, we introduce two separate MLP-based projection layers: one for the visual latents $v_{ok}^c$ and another for the diffusion timestep $k$. Both projections are mapped into the hidden dimension of the language backbone, and their outputs are summed before being fed into the backbone.

**Diffusion Forcing in Chunks** Autoregressive generation models are typically trained with teacher forcing, where the next token is predicted conditioned on the ground-truth history. While effective for discrete modalities such as text and speech, directly applying this scheme to continuous video latents often causes irreversible drift and frame corruption, due to the mismatch between training on ground-truth histories and inference on self-generated histories.

To achieve stable long-range video generation, we adopt diffusion-forcing Chen et al. (2024); Song et al. (2025b) for the video modality. Unlike standard diffusion models that apply a uniform noise level across all video tokens, we perturb each video chunk $v_o^c$ with an independent noise level $k^c$. All chunks are then trained to be denoised in parallel under noisy historical context. This design improves robustness against imperfect histories and effectively mitigates both inter-chunk and intra-chunk drift, ensuring coherent and consistent video generation over extended sequences.

**Global Identity Reference** While diffusion forcing alleviates error accumulation in video generation, maintaining long-range identity consistency remains challenging, directly impacting user immersion and interaction quality. Instead of relying on a heavyweight reference network to repeatedly inject identity features of $I_s$, we adopt a simpler yet effective approach: treating $i_s$ as a global condition and placing it at the start of the context sequence. This allows all generated video latents to consistently attend to the identity tokens. Notably, we observe that under this setup the model learns to balance identity cues dynamically, drawing from the global identity embedding while also leveraging historical context, resulting in outputs that are both coherent and identity-preserving.

**Real-Time Inference** The number of video tokens grows quadratically with spatial resolution. To balance real-time performance with the need for long-range visual context, we target 25-fps video synthesis at $256 \times 256$ resolution. However, even at this scale, the number of video tokens is $16\times$ greater than speech tokens, and unlike discrete token prediction which requires only a single model forward pass, each video token must undergo at least $N = 25$ denoising steps for stable generation.

For real-time streaming, we employ a standard Key–Value (KV) cache to avoid redundant computation during autoregressive generation. In addition, we introduce a chunk-wise pyramid denoising scheduler (detailed in the Appendix A.2) that significantly reduces the computational burden. Instead of requiring $|c| \times N$ forward passes for $|c|$ video chunks with $N$ denoising steps, our scheduler lowers this cost to $|c| + N - 1$, yielding a substantial speedup while preserving generation quality. Due to memory and latency constraints, we restrict the visual context to 2K tokens, corresponding to a 10-second window. Nevertheless, the conversational context remains unchanged in our Thinker as GLM-4-Voice supporting up to 8K tokens—roughly 10 minutes of dialogue. We do not apply Classifier-Free Guidance (i.e., CFG=1) for generation efficiency.

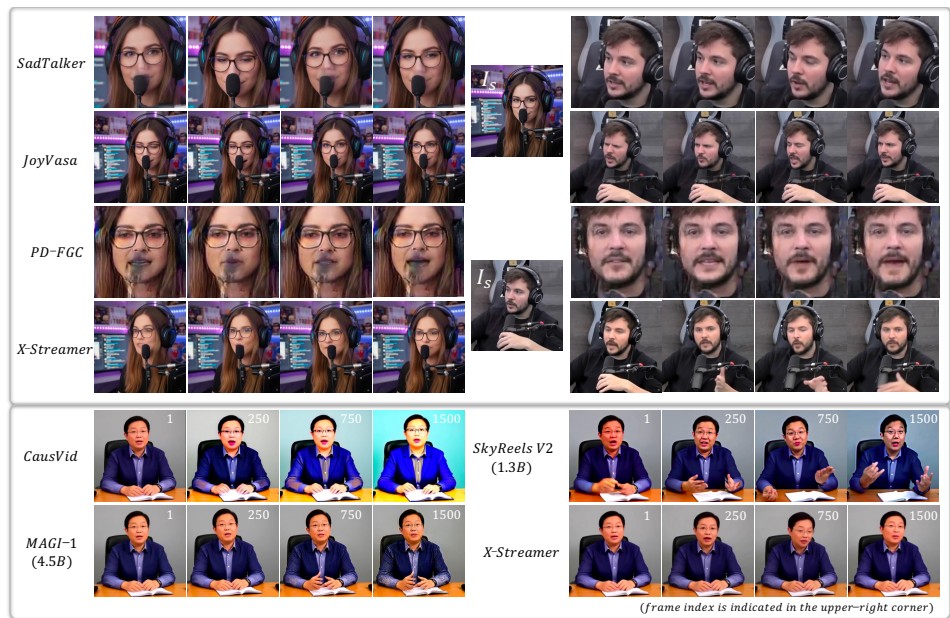

Figure 4: Qualitative comparisons on audio-synced (top) and long-range (bottom) video generations.

We build a real-time video call interface by distributing X-Streamer across two A100 GPUs, with the Thinker and Actor hosted separately. To ensure low-latency transmission between the remote GPU servers and client devices, we employ a cloud-based WebRTC service powered by LiveKit liv (2025). In Appendix A.3, we further demonstrate a straightforward extension of X-Streamer to support visual understanding of the user's video stream.

## 4 EXPERIMENTS

### 4.1 IMPLEMENTATION DETAILS

**Datasets.** We curate a large-scale corpus of talking-head videos by combining multiple public datasets—HDTF Zhang et al. (2021), CelebV-HQ Zhu et al. (2022)—together with additional licensed sources collected from online platforms. To ensure data quality, we apply a series of pre-processing and filtering steps, including scene-cut detection PySceneDetect (2025) and lip-sync validation Chung & Zisserman (2016), as detailed in Appendix A.1. The final dataset consists of approximately 2.7 million clips, totaling 4248.6 hours of footage, with an average duration of 5.5 seconds per clip. Each video is processed into multimodal triplets of text, speech, and video.

For evaluation, we assembled a benchmark of 50 in-the-wild human reference images collected from DeviantArt (2025), Midjourney (2025), and Pexels (2025), covering a wide range of identities, styles, and background contexts. In addition, we created a set of 50 multi-turn user queries, randomly generated using ChatGPT, to assess extended conversational robustness.

**Training.** We leverage the pretrained conversational intelligence of GLM-4-Voice and train only the Actor's video transformer on our multimodal sequences. During training, text and audio streams are processed by the Thinker, whose hidden states guide the learning of the video modality. Training proceeds in two stages. In pretraining, we use 2.7M clips of 5–20 seconds, training for 3 epochs on 256 A100 GPUs with AdamW, a per-GPU batch size of 2, and a learning rate of $1 \times 10^{-5}$. In finetuning, we train on 220K high-quality long-form samples for 200K steps using the same learning rate. We do not apply instruction finetuning to the full model, as synthetic QA pairs derived from talking-head transcripts lack sufficient quality and depth. Instead, at inference time, GLM-4-Voice handles text and speech generation, while the Actor specializes in translating its hidden states into synchronized multimodal streams.

**Inference.** The video stream is generated in 8-frame chunks (64 video tokens), yielding 384 video latents (6 chunks) per multimodal segment interleaved with 13 text tokens and 26 speech tokens. This setup ensures that video synthesis is fully guided by text and speech outputs. On a single GPU, the full model peaks at 53 GB of VRAM. However, generating a 1-minute multimodal response

Table 1: Quantitative evaluation. The **best** and second-best scores are highlighted.

| Method | CPBD↑ | FVD↓ | ID-Sim↑ | SynC↑ | SynD↓ | Glo↑ | Exp↑ | ID↑ | Lip↑ | Div↑ | VQ↑ |
|---|---|---|---|---|---|---|---|---|---|---|---|
| JoyVasa | 0.37 | 748.99 | 0.73 | 2.84 | 11.10 | 0.03 | 0.021 | 0.1 | 0.13 | 0.08 | 0.08 |
| SadTalker | 0.20 | 777.52 | **0.78** | 3.39 | 11.15 | 0.04 | **0.035** | 0.13 | 0.08 | 0.05 | 0.06 |
| PD-FGC | 0.17 | 1183.82 | 0.42 | **4.22** | 11.20 | 0.003 | 0.008 | 0 | 0.09 | 0.02 | 0 |
| **X-Streamer** | **0.55** | **573.36** | 0.75 | 3.41 | **10.93** | **0.081** | 0.033 | **0.77** | **0.7** | **0.85** | **0.86** |

Table 2: Qualitative Ablation. The **best** and second-best scores are highlighted.

| Method | CPBD ↑ | FVD ↓ | ID-Sim ↑ | Glo ↑ | Exp ↑ |
|---|---|---|---|---|---|
| w/o diffusion forcing | 0.17 | 1989.52 | 0.29 | **0.246** | 0.012 |
| w/o global ID ref | 0.26 | 794.78 | 0.41 | 0.053 | 0.02 |
| token-wise causal attn | 0.37 | 628.76 | 0.70 | 0.035 | 0.023 |
| **X-Streamer** | **0.55** | **573.36** | **0.7542** | 0.081 | **0.033** |

takes 51.2 seconds and 58.2 seconds for the Thinker and Actor respectively. To overcome this, we distribute the dual transformers across two A100 GPUs, achieving 25 fps multimodal streaming.

## 4.2 EVALUATION

**Baselines.** Real-time audiovisual interaction remains underexplored Ao (2024); Low & Wang (2025); Zhu et al. (2025); Chen et al. (2025c); Wang et al. (2025b), with no open-source methods currently available. We therefore compare X-Streamer against representative *real-time audio-driven* portrait animation work: JoyVasa Cao et al. (2024), an open-source implementation of VASA-1 Xu et al. (2024b); SadTalker Zhang et al. (2023), a GAN-based method decoding from implicit facial motion latents; and PD-FGC Wang et al. (2023), which offers disentangled control over lip motion and facial expressions. For fairness, all baselines are driven by audio synthesized with X-Streamer, and evaluations are conducted on generated videos at a fixed resolution of $256 \times 256$.

Real-time video streaming remains underexplored Yin et al. (2025); Lin et al. (2025b); Kodaira et al. (2025); Huang et al. (2025b). We compare our model against the publicly available method of Yin et al. (2025). The Self Forcing approach Huang et al. (2025b) is excluded, as its released model supports only short generations under 10 seconds, whereas our setting requires sustained interaction lasting minutes to hours. We also include SkyReels-V2 (1.3B) Chen et al. (2025a) and MAGI-1 (4.5B) Teng et al. (2025) as autoregressive video diffusion baselines, though they require hours to synthesize a single one-minute video. Notably, these baselines are neither audio-conditioned nor capable of generating audio; for fairness, we condition their video outputs on a fixed text prompt.

**Qualitative Evaluation.** Qualitative comparisons between X-Streamer and baseline methods are presented in Fig.4. X-Streamer generalizes well to chest-level portraits and remains robust under occlusion, side views, and complex environments, producing dynamic and natural motions. In contrast, SadTalkerZhang et al. (2023) and PD-FGC Wang et al. (2023) focus narrowly on facial regions and often exhibit artifacts when the face is partially occluded (e.g., the microphone obscuring the mouth in the left example). JoyVasa Cao et al. (2024) shows stronger robustness but generates motion that is relatively rigid and constrained, whereas X-Streamer produces coordinated head movements and expressive hand gestures, yielding more lifelike interactions. We further compare with CausVid Yin et al. (2025) (bottom rows in Fig. 4) to evaluate stability in long-horizon video streaming. CausVid remains stable for the first few seconds, but its spatial fidelity and identity consistency degrade noticeably after around 10 seconds. Similarly, SkyReels-V2 and MAGI-1, though running offline, suffer from identity drift and color inconsistencies within 30 seconds. In contrast, X-Streamer maintains temporally stable generation with consistent identity throughout the entire sequence.

**Quantitative Evaluation.** We evaluate X-Streamer against real-time audio-driven portrait animation baselines using metrics that assess visual fidelity, identity preservation, audiovisual synchronization, and temporal dynamics (Table 1). Visual quality is measured with Cumulative Probability of Blur Detection (CPBD↑ Narvekar & Karam (2011)) and Fréchet Video Distance (FVD↓ Unterthiner et al. (2019)). Identity consistency is quantified by cosine similarity of ArcFace embeddings Deng et al. (2019), reported as ID-Sim↑. Audiovisual alignment is evaluated with SynC↑ and SynD↓ Chung & Zisserman (2016), which measure speech–lip synchronization. Naturalistic dynamics are captured with Global Motion (Glo↑) and Dynamic Expression (Exp↑), quantifying head motion and upper-face expressions while excluding the mouth region. In addition to objective metrics, we conduct a user study (20 participants, 100 choices per dimension) comparing our method

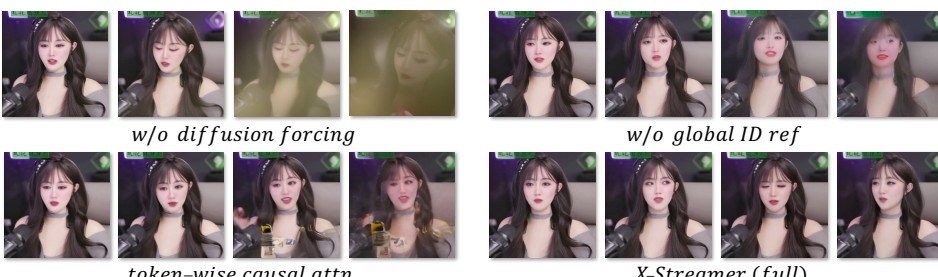

*w/o diffusion forcing*          *w/o global ID ref*

*token–wise causal attn*          *X-Streamer (full)*

Figure 5: **Visual Ablation.** Diffusion forcing and global identity referencing stabilize long-horizon video generation, while applying spatially bidirectional attention within each video chunk (as opposed to fully causal token-wise attention) reduces flickering and preserves structural integrity.

and baselines across four aspects: identity preservation (ID↑), lip synchronization (Lip↑), motion diversity (Div↑), and overall video quality (VQ↑).

As shown in Table 1, our method outperforms all baselines in visual fidelity (CPBD↑ Narvekar & Karam (2011), FVD↓ Unterthiner et al. (2019), and VQ↑) as well as motion dynamics (Glo↑ and Div↑). While X-Streamer ranks second in objective ID similarity (ID-Sim↑) due to SadTalker's restricted motion and zoomed-in facial framing, the user study highlights X-Streamer's superior identity preservation (ID↑). Our approach also demonstrates strong lip synchronization, achieving the lowest SynD↓ alongside superior Lip↑ scores.

### 4.3 ABLATION STUDY

We ablate key components of our framework by replacing them with alternative designs and evaluating on the test set. Quantitative results are reported in Table 2, with visual comparisons in Figure 5 and on our supplementary webpage. Replacing diffusion forcing with standard teacher forcing causes prediction errors to accumulate, leading to motion drift and degraded visual quality. This variant shows the highest Glo↑ due to undesirable motion artifacts, consistent with its lowest FVD↓. Removing the global identity reference forces the model to rely sorely on visual history, which leads to facial distortions and color drift in long-horizon sequences, as reflected in lower ID-Sim↑. Finally, replacing our spatiotemporal attention design (temporal-causal with spatially bidirectional attention) with fully causal token-wise attention reduces temporal coherence and weakens visual fidelity, lowering CPBD↑ and worsening FVD↓. Together, these results confirm that our full model achieves stable long-duration video streaming with strong fidelity and identity consistency.

## 5 CONCLUSION

We introduced X-Streamer, an end-to-end multimodal interactive human world modeling framework that unifies text, speech, and video understanding and generation within a single architecture. At its core, we proposed a Thinker–Actor dual-transformer design: the Thinker performs conversational reasoning, while the Actor converts its hidden states into synchronized, streaming multimodal responses. Extending language models to the video modality with chunk-wise diffusion forcing, our framework balances real-time efficiency, long-range consistency, and temporal multimodal synchronization. Extensive experiments demonstrate that X-Streamer is a significant step toward persistent, interactive, and intelligent digital humans and world modeling.

**Limitation and Future Work.** X-Streamer extends a language–speech model to the video modality but is trained solely on real-human talking-head videos, limiting its generalization to broader scenarios. Since our framework is orthogonal to the backbone choice, it can naturally benefit from future advances in language–speech models, yielding richer voices, emotions, and expressiveness. Higher-resolution, real-time video generation with ultra-long audiovisual context is also feasible through fewer-step distillation Yin et al. (2025); Lin et al. (2025b); Huang et al. (2025b) and advanced context management Cai et al. (2025); Guo et al. (2025); Zhang & Agrawala (2025), which we leave for future work. Beyond conversational interactions, an important direction is to expand X-Streamer toward broader multimodal engagement, such as perceiving the user's video stream (Appendix A.3), interacting with objects, and following multimodal commands. Addressing these challenges will move X-Streamer closer to a general-purpose world modeling framework for digital humans, enabling open-ended, context-aware interaction.

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

# A APPENDIX

## A.1 DATASET

We curate a large-scale bilingual (English and Chinese) audio–visual pretraining corpus We curate a large-scale bilingual (English and Chinese) audio–visual pretraining corpus comprising 2,780,920 samples with a total duration of 4,248.6 hours. Other languages are excluded since they are not supported by the GLM-4-Voice speech tokenizer. All source videos are processed automatically using our pipeline (detailed below). To ensure accurate audio–visual alignment, we retain only those segments whose lip-sync score, measured by SyncNet Chung & Zisserman (2016), exceeds 3.5. On top of this corpus, we construct a supervised fine-tuning (SFT) subset containing 217,074 samples (331.64 hours). Within this subset, 5,406 samples (about 62 hours, with an average length of 41 seconds per sample) are longer than 20 seconds. These are selected under strict quality criteria, including an image quality (IQA) Su et al. (2021) score of at least 70, exactly one detected speaker, no optical character recognition (OCR)-detected overlays, and a SyncNet score of at least 5.0.

The data curation pipeline follows a fixed, carefully designed sequence to ensure temporal consistency, visual integrity, and precise audio–visual alignment. First, scene detection segments long videos into coherent shots PySceneDetect (2025). Lip-sync filtering is then applied to remove poorly aligned clips, using scores computed by SyncNet Chung & Zisserman (2016). Human detection guarantees that each segment contains exactly one visible speaker Ge et al. (2021), while face detection and tracking maintain accurate localization and identity continuity across frames. Next, aesthetic filtering removes visually low-quality shots Su et al. (2021), and OCR-based screening

eliminates samples with overlays or watermarks EasyOC (2025). The audio track is subsequently denoised to suppress background noise Rouard et al. (2022). Automatic speech recognition (ASR) is then performed to obtain transcripts, which are later used as an auxiliary conditioning stream for modeling Radford et al. (2022); Gao et al. (2023). Finally, modality-specific embeddings for text, audio, and video are precomputed and cached Zeng et al. (2024); HaCohen et al. (2024), reducing I/O and preprocessing overhead during training. This tightly integrated pipeline produces a clean, large-scale dataset that supports reliable and efficient audio–visual learning and alignment modeling.

## A.2 DIFFUSION FORCING DENOISING SCHEDULING

$$
\mathcal{K}^{chunk} = \begin{bmatrix}
N & \cdots & N & N & \cdots & N & \cdots & N & \cdots & N \\
N\text{-}1 & \cdots & N\text{-}1 & N & \cdots & N & \cdots & N & \cdots & N \\
N\text{-}2 & \cdots & N\text{-}2 & N\text{-}1 & \cdots & N\text{-}1 & & N & \cdots & N \\
\vdots & & \vdots & \vdots & & \vdots & \ddots & \vdots & \cdots & \vdots \\
1 & \cdots & 1 & 2 & \cdots & 2 & \cdots & c & \cdots & c \\
0 & \cdots & 0 & 1 & \cdots & 1 & \cdots & c\text{-}1 & \cdots & c\text{-}1 \\
\vdots & & \vdots & \vdots & & \vdots & \ddots & \vdots & \cdots & \vdots \\
0 & \cdots & 0 & 0 & \cdots & 0 & \cdots & 1 & \cdots & 1 \\
0 & \cdots & 0 & 0 & \cdots & 0 & \cdots & 0 & \cdots & 0
\end{bmatrix}
$$

$\overbrace{\hspace{2cm}}^{1st\ chunk} \quad \overbrace{\hspace{2cm}}^{2nd\ chunk} \quad \overbrace{\hspace{2cm}}^{c^{th} chunk}$

Figure 6: Scheduler $\mathcal{K}^{chunk}$ of chunk-wise pyramid denoising.

Diffusion forcing Chen et al. (2024) organizes the denoising schedule for each latent through a scheduling matrix $\mathcal{K}$. Building on this idea, we propose a chunk-wise pyramid variant, $\mathcal{K}^{chunk}$, where denoising proceeds sequentially across chunks. This design enables chunk-level parallelism during inference and reduces the number of forward passes from the conventional $|c| \times N$ required by a chunk-by-chunk DDIM scheduler to $|c| + N - 1$, where $|c|$ is the number of chunks and $N$ the number of denoising steps. An illustration of $\mathcal{K}^{chunk}$ is shown in Fig. 7. Each row in the matrix specifies the noise level assigned to tokens at a given denoising round. Starting from a fully noised sequence (top row), the algorithm progressively denoises chunks in order, refining their latent representations. The height of $\mathcal{K}^{chunk}$ thus corresponds to the total number of forward passes needed to generate the full sequence.

## A.3 EXTENDING X-STREAMER WITH VISUAL PERCEPTION

Since GLM-4-Voice does not process visual inputs, the current system is limited to user queries in text and speech. To illustrate how X-Streamer can be extended with perception capabilities, we incorporate an auxiliary visual–language model (VLM) to analyze webcam video streams. As shown in Fig. 7, when a user issues a query, the VLM processes the latest video frame and produces a concise textual description of the scene. This description is injected into the Thinker's input prompt, enabling the model to reason over both user queries and up-to-date visual context. This extension grounds X-Streamer's responses in live visual evidence without requiring manual annotations, demonstrating how the framework can integrate perception with multimodal generation for richer, context-aware interaction.

## A.4 MORE RESULTS

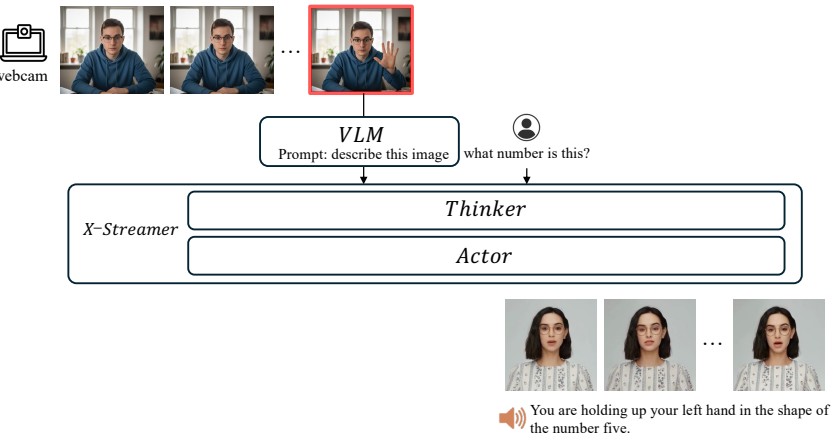

Figure 7: X-Streamer with visual perception. When the user issues a query (e.g., "what number is this?"), a VLM analyzes the current webcam frame and produces a concise textual description, which is passed to X-Streamer to guide subsequent multimodal response generation.

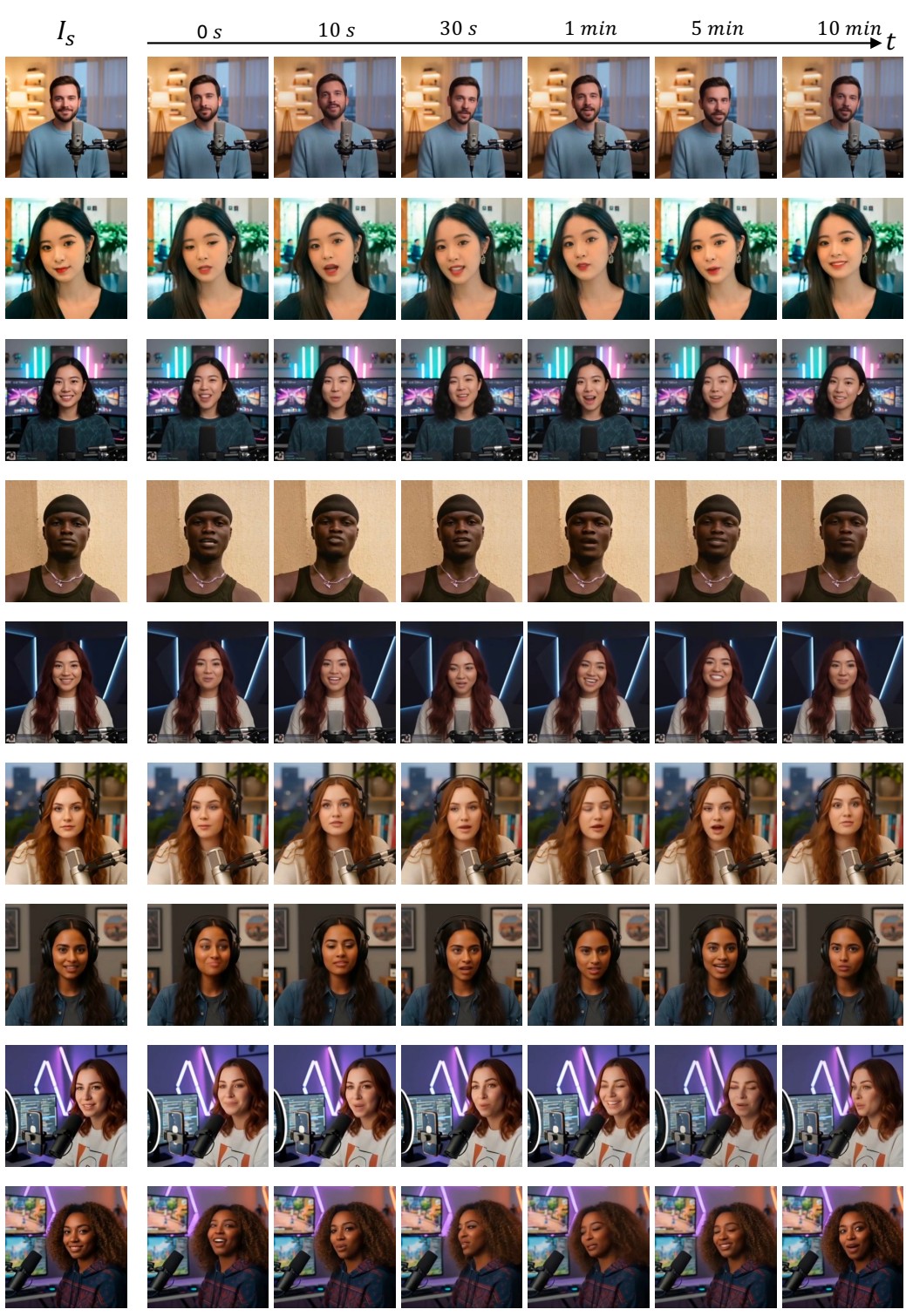

Figure 8: More results of X-Streamer.

