# OpenReview forum: "X-Streamer: Unified Human World Modeling with Audiovisual Interaction"
_ICLR.cc/2026/Conference — Submitted to ICLR 2026_

### Official Review · Reviewer_EKMn · 2025-10-31

**Soundness:** 3
**Presentation:** 3
**Contribution:** 3
**Rating:** 6
**Confidence:** 3

**Summary:**

The paper proposes a **dual-transformer architecture** to achieve unified multimodal understanding and generation. Specifically, the **VLM branch** handles comprehension and produces text and audio outputs, while the **diffusion branch** generates video. The two branches are temporally and semantically aligned via **cross-attention**. Additionally, the authors introduce **inter- and intra-chunk attentions** and **time-aligned multimodal positional embeddings** to further strengthen temporal and semantic coherence across modalities.

**Strengths:**

- The paper is **well-structured and clearly written**, with a coherent presentation of ideas.
- It makes a **notable contribution** by introducing a unified framework for interactive human agents, which was previously handled by modular or pipeline-based systems.
- The proposed approach shows **significant improvements** in **modality alignment**, **temporal consistency**, and **generation efficiency**, representing a meaningful step forward for multimodal generation systems.

**Weaknesses:**

1. According to the description, the full system seems to include **Qwen-Omni’s Talker** for audio decoding, but this component is **not reflected in Figure 2**. The authors should clarify its role and integration.
2. As the autoregressive (AR) generation progresses, the **diffusion model’s context length** naturally grows, which could impact decoding speed. Is there **context truncation** during deployment, and if so, **how is it implemented**? Does truncation affect temporal or semantic consistency?
3. Could a **non-autoregressive diffusion model** serve as a viable baseline for video decoding? For instance, such a model could take as input a portrait, short text, and audio segments to generate short video clips in parallel. Since the portrait remains a consistent conditioning signal, **identity preservation** might still be maintained. The paper does not seem to include a comparison to this baseline.

**Questions:**

See weakness above

---

> ### Author Response · Authors · 2025-11-18
> **Author Response to Reviewer EKMn**
>
> We sincerely thank Reviewer EKMn for the insightful comments and questions, and we appreciate the recognition of our work as **"a notable contribution"** toward **"a unified framework for interactive human agents."** Below we provide detailed responses to each of the reviewer’s questions.
> ***
> **1. Audio Decoder.**
>
> Our design is inspired by Qwen-Omni’s Thinker–Talker generative paradigm, but we do **NOT** use Qwen-Omni's audio decoding modules. Instead, our system relies on the built-in audio tokenizer and decoder of GLM-4-Voice, as described in Sec. 3.1.
>
> **2. Context Management.**
>
> We agree with the reviewer that the context naturally grows as autoregressive generation proceeds. To ensure real-time memory usage and latency, we employ a context-management strategy with sliding-window truncation, as described in Sec. 3.1 (“Audio-Visual Context Attentions”) and Sec. 3.2 (“Real-Time Inference”).
>
> For the Actor module, we apply a sliding-window self-attention over visual tokens together with a constant global reference. We cap the visual context at 2K tokens, corresponding approximately to the most recent 10 seconds of visual history. In contrast, the conversational context is preserved within the Thinker module, where GLM-4-Voice supports up to 8K tokens (≈10 minutes of dialogue), and this long-range text/audio context cross-attends to the visual tokens in the Actor.
>
> Empirically, this audio–visual context design maintains both semantic consistency and pixel continuity, while satisfying the real-time memory and latency constraints required for our system, as demonstrated in the provided demos.
>
> **3. A Modular Baseline: Text/Audio-Driven Short Video Clip Generation.**
>
> We thank the reviewer for suggesting this baseline, which corresponds to a modular pipeline where the Actor functions as a text/audio-conditioned video diffusion model that generates very short video chunks (to meet real-time requirements) in a non-autoregressive manner. We are happy to include this baseline in the revised version. Below we highlight two key differences from our proposed approach:
>
> (1) **Latency.** This modular pipeline requires full-chunk text/audio decoding prior to video generation, which introduces non-negligible cross-modal latency. In contrast, our method performs interleaved, synchronized generation across text, audio, and video, ensuring tightly aligned multimodal streaming.
>
> (2) **Visual discontinuity.** Even if the baseline were to use the Thinker's audio-textual hidden states as we proposed in this work, it falls into a special case of our method with zero visual history, aside from a single global identity reference. This inevitably leads to chunked motion, noticeable temporal discontinuities, and a strong drift back toward the reference image, producing rigid or jittery motions. Maintaining continuous visual history is essential for preserving both visual semantics and pixel-level continuity, justifying the necessity of autoregressive video generation for real-time streaming.

---

> > ### Author Response · Authors · 2025-11-24
> > **Follow Up on Our Response**
> >
> > Dear Reviewer EKMn,
> >
> > Thank you again for your detailed and insightful comments on our submission. We wanted to follow up to check whether our responses have sufficiently clarified the questions you raised, or if there are any points on which further elaboration would be helpful. Many thanks again for your thoughtful review and your time.
> >
> > Best regards,
> >
> > Authors

---

### Official Review · Reviewer_J6da · 2025-10-31

**Soundness:** 3
**Presentation:** 2
**Contribution:** 2
**Rating:** 4
**Confidence:** 3

**Summary:**

This paper introduces X-Streamer, an end-to-end multimodal framework for building interactive digital humans that enable real-time text-speech-video interactions from a single input portrait. It adopts a Thinker–Actor dual-transformer architecture, where the frozen Thinker interprets streaming text/audio inputs, and the Actor autoregressively generates interleaved multimodal outputs. The quantitative and qualitative results show that X-Streamer achieves strong and consistent performance in visual quality, temporal coherence, and audio–visual synchronization during real-time talking-head interactions.

**Strengths:**

- The paper introduces a unified architecture that seamlessly integrates text, speech, and video generation for real-time digital humans.
- X-Streamer achieves strong visual quality and maintains real-time performance.
- The paper shows stable and visually coherent long-form video generation, maintaining identity and temporal consistency over minutes of continuous interaction.

**Weaknesses:**

1. Limited novelty: The approach primarily constitutes an engineering integration of existing components (e.g., GLM‑4‑Voice backbone, diffusion forcing) and introduces few fundamentally new modeling ideas.
2. Insufficient experiments: The experiments on long-duration videos are limited; more cases and in-depth analyses are needed. In addition, ablation studies should examine factors such as the impact of visual context length on both performance and latency.
3. Limited scope of contribution: Although the paper frames its contribution as “human world modeling,” the demonstrated capabilities are largely restricted to audiovisual talking-head generation, which slightly overstates the scope of the claimed modeling.

**Questions:**

1. Please refer to points listed in weakness.
2. How does the visual context length affect performance and latency? Did the authors explore trade-offs between longer visual context for identity preservation and real-time efficiency?
3. Would jointly fine-tuning the Thinker module with the Actor improve long-term temporal consistency, multimodal alignment, or identity preservation compared to keeping it frozen?
4. Table 2 claims qualitative ablation in capiton while the content are quantitative results.

---

> ### Author Response · Authors · 2025-11-18
> **Author Response to Reviewer J6da**
>
> We sincerely thank Reviewer J6da for the constructive feedback and insightful questions. We also appreciate the reviewer's recognition of the **"strong visual quality"** and **"real-time performance"** achieved by our method for **"stable and visually coherent long-form video generation."**  Below we provide detailed responses to each of the reviewer’s questions.
> ***
> **1. Long-Duration Results and Visual Context Length Analyses.**
>
> We have included real-time recorded long-duration interaction results on the supplementary webpage, where we engage with the agent continuously for 50 minutes (the playback is accelerated by 16× to avoid an overlong demo). In the revision, we will provide additional examples to further demonstrate the infinite streaming capability of our method.
>
> To quantitatively support long-duration consistency, we conducted additional identity-consistency evaluations. Using ArcFace similarity [1], we observe that identity preservation remains consistently high throughout the entire streaming session: the average similarity stays around 0.7315, with 0.7845 at 1 minute, 0.7187 at 20 minutes, 0.7510 at 40 minutes, and 0.7293 at 50 minutes. These results numerically confirm our method’s ability to maintain long-horizon visual and identity consistency during extended real-time streaming.
>
> We further performed ablation studies on visual context length. Our default configuration uses a sliding-window visual context of 2K tokens (approximately 10 seconds of recent visual history). Doubling the visual context length substantially increases resource cost: VRAM peak rises from 53GB to 73GB, and generation time for a 2-second clip increases from 1.78s to 2.26s on average, while providing no improvement in identity preservation (ArcFace similarity changes from 0.7500 to 0.7423). Conversely, reducing the visual context window degrades semantic consistency and leads to significantly more rigid motion and frequent resets toward the reference image, due to the dominance of the global identity embedding. These findings show that our chosen context length offers the best balance between identity consistency, motion fluidity, latency, and memory footprint. We will include these ablations in the revised version.
>
> **2. Joint Fine-Tuning.**
>
> As described in Sec. 4.1 (“Training”), we train the dual-transformer architecture end-to-end while keeping the Thinker (based on GLM-4-Voice) frozen to fully leverage its pretrained audio–text conversational capabilities. During training, we update the Actor and the connector that maps the Thinker’s hidden states into the visual modality.
>
> We also experimented with joint finetuning of the Thinker on our talking-head audio–video data. However, this led to a substantial degradation in the Thinker’s conversational QA ability. The main reason is the lack of meaningful conversational supervision in talking-head datasets: obtaining high-quality real-world omni-modal training tuples, i.e., text/audio questions paired with synchronized audio/video answers, is extremely challenging. Meanwhile, synthetic QA pairs derived from transcripts lack the depth and semantic richness needed to preserve the Thinker’s reasoning skills.
>
> Therefore, freezing the Thinker while training the Actor and connector provides an effective and cost-efficient strategy for building intelligent conversational agents spanning text, audio, and video modalities.

---

> > ### Author Response · Authors · 2025-11-18
> > **Author Response to Reviewer J6da**
> >
> > **3. Novelty.**
> >
> > We respectfully disagree with the characterization that the work lacks novelty. The contribution of X-Streamer is not in introducing yet another backbone model, but in the architectural and algorithmic advances required to make long-horizon, real-time multimodal streaming interactions across text, audio, and video actually work, a capability that current multimodal LLMs and video generation systems fundamentally do not support.
> >
> > Below we highlight the three key technical contributions of our work:
> >
> > First, we propose a unified generative paradigm for multimodal, multi-turn interactions via a two-level Thinker–Actor architecture. This design explicitly decouples long-horizon semantic reasoning from low-level visual synthesis. The Thinker operates fully in token space to plan multimodal semantics (content, prosody, emotion), while the Actor realizes these plans in spatiotemporal space through diffusion decoding. This separation enables long-range omni-modal consistency, which cannot be achieved by monolithic one-pass models or standard LLM-conditioned pipelines. Our formulation also serves as a pioneering step toward general human world modeling.
> >
> > Second, at the Actor level, we enable unbounded, real-time, and stable video streaming with tight cross-modality synchronization, which to our knowledge is unprecedented in prior work. We design an autoregressive diffusion transformer operating in compressed video-VAE latent space, reinforced by diffusion forcing and a global identity anchor to maintain identity and stability over long horizons. Moreover, we carefully determine the multimodal segment duration and introduce a chunk-wise cross-attention mechanism to achieve accurate audio–lip synchronization while preserving long-range temporal coherence. These components have been extensively ablated and are all critical to the system's performance.
> >
> > Third, we demonstrate a fully functional real-time interactive system capable of long-horizon audio–visual streaming and interaction. Delivering this system requires substantial algorithmic and engineering innovations, including considerations around latency, consistency, synchronization, and visual fidelity. Importantly, our framework provides a cost-efficient training paradigm for extending pretrained language–speech models to the video domain: it requires only talking-head videos, without the need for expensive omni-modal triplets, yet still achieves strong cross-modal alignment in real-time settings.
> >
> > Demonstrating a system with these capabilities is nontrivial and constitutes a substantive research contribution, particularly in a field where streaming generation, digital humans, and world-model agents are increasingly evaluated by their real-world applicability.
> >
> > **4. Human World Modeling: Problem Formulation and Scope.**
> >
> > In this work, we introduce a problem formulation for human world modeling and envision a new generative paradigm for digital human agents built around two key capabilities: (1) infinite streaming multimodal interaction with retention of long-range, multi-turn context, and (2) first-person audiovisual engagement through intelligent, context-aware responses.
> >
> > Within this broader vision, we focus on a specific subdomain of human world modeling, long-range audio–visual interaction for portrait-scale agents, which has remained largely underexplored prior to our work. Rather than claiming to solve the full problem, we view our method as a pioneering step toward a unified and general multimodal human world modeling framework. As discussed in Sec. 5 ("Limitation and Future Work"), extending our approach to broader forms of multimodal engagement is an important and promising direction for future research.
> >
> > In the revision, we will further clarify the problem scope addressed by this work and explicitly tone down broader claims, highlighting instead how our formulation aims to motivate and inspire subsequent research toward general human world modeling.
> >
> > **5. Typo.**
> >
> > Thank you for pointing it out and we will fix it in our revision.
> > ***
> > [1] Jiankang Deng, Jia Guo, Niannan Xue, and Stefanos Zafeiriou. Arcface: Additive angular margin loss for deep face recognition. In Proceedings of the IEEE/CVF conference on computer vision and pattern recognition, pp. 4690–4699, 2019.

---

> > > ### Author Response · Authors · 2025-11-24
> > > **Follow Up on Our Response**
> > >
> > > Dear Reviewer J6da,
> > >
> > > Thank you again for your detailed and insightful comments on our submission. We submitted our rebuttal last week addressing each of the points you raised, and we would be grateful to know whether our explanations have resolved your concerns.If any aspects remain unclear, we would be happy to provide additional experimental details or extended results to further support our claims. We sincerely appreciate your time and effort in reviewing our work.
> > >
> > > Warm regards,
> > >
> > > Authors

---

> > > ### Comment · Reviewer_J6da · 2025-11-28
> > >
> > > Thank you for the clarifications. The rebuttal does address some of my earlier questions regarding results of long duration videos and influence of joint-tuning, and I appreciate the additional information. However, my principal concern remains unresolved: the manuscript still does not clearly establish a scientific contribution beyond system integration.
> > >
> > > 1. The paper frames a broad product- and system-level objective, rather than a clearly defined research problem.
> > > While the manuscript outlines high-level challenges associated with building a real-time digital human system, it does not distill these into a precise scientific question that the proposed architecture uniquely addresses. As a result, the contribution is presented as progress toward an application-driven system goal rather than a focused research advance.
> > >
> > > 2. Most core components closely follow established practices.
> > > The Thinker–Actor mechanism, chunk-wise generation, diffusion forcing, etc closely follow patterns in previous works[1,2,3]. The identity reference is also commonly used in various multimodal tasks, such as image-to-video[4], image editing[5]. Their combination is technically strong but does not constitute a new modeling paradigm.
> > >
> > > **Regarding the scope of the proposed methods**, the authors acknowledge in the rebuttal that while they introduce the broader notion of human world modeling, the actual method targets only a specific subdomain—portrait-scale audio–visual interaction with a fixed viewpoint and constrained dynamics. Since the paper does not address the wider aspects typically associated with human world modeling, framing the work under this broader term in the title overstates the scope and may be misleading for readers.
> > >
> > > In summary, while the authors have resolved some implementation concerns, the manuscript still does not present a sufficiently clear or novel scientific contribution for ICLR. The central issues concerning problem formulation, novelty, and generality remain unaddressed.
> > >
> > > [1] Xu, Jin, et al. "Qwen2. 5-omni technical report." arXiv preprint arXiv:2503.20215 (2025).
> > >
> > > [2] Teng, Hansi, et al. "MAGI-1: Autoregressive Video Generation at Scale." arXiv preprint arXiv:2505.13211 (2025).
> > >
> > > [3] Chen, Boyuan, et al. "Diffusion forcing: Next-token prediction meets full-sequence diffusion." Advances in Neural Information Processing Systems 37 (2024): 24081-24125.
> > >
> > > [4] Blattmann, Andreas, et al. "Stable video diffusion: Scaling latent video diffusion models to large datasets." arXiv preprint arXiv:2311.15127 (2023).
> > >
> > > [5] Liu, Shiyu, et al. "Step1x-edit: A practical framework for general image editing." arXiv preprint arXiv:2504.17761 (2025).

---

> > > > ### Author Response · Authors · 2025-12-01
> > > > **Author Response to Reviewer J6da**
> > > >
> > > > We appreciate that our last response has clarified some points regarding the long-duration results and the influence of joint fine-tuning. However, we respectfully but firmly disagree with the reviewer's remaining claims, which are based on a mismatch between the assumed capabilities of prior systems and what they actually support.
> > > >
> > > > **1. Our paper addresses a well-defined sequence modeling problem.**
> > > >
> > > > The central question of our paper is: *"How to achieve unified autoregressive text–audio–video generation with real-time latency, unbounded temporal horizon, and cross-modal semantic/identity/prosody consistency".*
> > > >
> > > > Eq. (1)–(2) formalize this as a multimodal autoregressive process requiring **persistent cross-modal semantic states** and **synchronized multimodal token generation**—a structure absent in prior multimodal architectures. Our work provides one of the first modeling formulations for this problem,  which is fundamentally different from a system-level goal. Characterizing this as such overlooks both the inherent complexity and the significance of enabling real-time multimodal autoregression.
> > > >
> > > > **2. Our approach is not "system integration" and no prior method supports real-time, long-horizon multimodal autoregression.**
> > > >
> > > > None of the reference works [1-5] provide:
> > > >
> > > > - unified autoregressive **text–audio–video** generation,
> > > >
> > > > - **real-time** operation,
> > > >
> > > > - **unbounded** temporal horizon,
> > > >
> > > > - **stable multimodal coherence** over tens of minutes.
> > > >
> > > > Qwen2.5-Omni [1] adopts a Thinker–Talker design where text generation with Thinker is decoupled from speech generation with Talker and does not perform streaming video autoregression. In contrast, our Thinker–Actor architecture is fundamentally different:
> > > >
> > > > - The Thinker performs *long-horizon semantic planning* entirely in token space,
> > > >
> > > > - the Actor leverages the Thinker’s semantic states to *generate text, audio, and video tokens in an interleaved and temporally aligned manner*, enabling fully synchronized multimodal autoregression.
> > > >
> > > > This directly targets the three key challenges explicitly highlighted by Qwen2.5-Omni: inter-modal interference control, cross-modal temporal alignment, and real-time performance. None of the cited papers deliver a functioning long-horizon audio–image–video streaming agent.
> > > >
> > > > MAGI-1 [2] performs offline video generation and exhibits rapid identity drift (Fig. 4); Diffusion Forcing [3] is a training strategy; SVD [4] and Step1X-Edit [5] target single-image/video generation. These do not operate in the regime we study.
> > > >
> > > > **3. Scope clarification is acknowledged and will be refined.**
> > > >
> > > > Our work explicitly focuses on the portrait-scale, long-horizon multimodal streaming setting (Abstract; Sec. 1, 3). Broader human world modeling is discussed only as future work (Sec. 5). We will revise the wording to avoid overclaiming while clearly positioning our method as a first modeling step toward this broader direction.
> > > >
> > > > **Summary**
> > > >
> > > > Our contribution is not system integration, but a **new modeling principle—absent in prior work**: a hierarchical semantic planner coupled with a synchronized multimodal actor, enabling the **first real-time, unbounded-horizon autoregressive text–audio–video generation system.** The concerns rely on treating our approach as equivalent to prior work, whereas those methods target fundamentally different modeling regimes.
> > > >
> > > > [1] Xu, Jin, et al. "Qwen2. 5-omni technical report." arXiv preprint arXiv:2503.20215 (2025).
> > > >
> > > > [2] Teng, Hansi, et al. "MAGI-1: Autoregressive Video Generation at Scale." arXiv preprint arXiv:2505.13211 (2025).
> > > >
> > > > [3] Chen, Boyuan, et al. "Diffusion forcing: Next-token prediction meets full-sequence diffusion." Advances in Neural Information Processing Systems 37 (2024): 24081-24125.
> > > >
> > > > [4] Blattmann, Andreas, et al. "Stable video diffusion: Scaling latent video diffusion models to large datasets." arXiv preprint arXiv:2311.15127 (2023).
> > > >
> > > > [5] Liu, Shiyu, et al. "Step1x-edit: A practical framework for general image editing." arXiv preprint arXiv:2504.17761 (2025).

---

### Official Review · Reviewer_16Le · 2025-10-31

**Soundness:** 2
**Presentation:** 3
**Contribution:** 2
**Rating:** 4
**Confidence:** 3

**Summary:**

The paper introduces X-Streamer, an end-to-end multimodal framework that generates synchronized text, speech, and video streams from a single portrait for digital human interactions. The framework is built upon a Thinker–Actor dual-transformer architecture. A frozen pretrained language–speech model acts as the Thinker for perception and reasoning. An Actor autoregressively generates synchronized multimodal outputs using chunk-wise diffusion forcing and global identity referencing. The framework runs in real time on two A100 GPUs, sustaining hours-long consistent video chat experiences.

**Strengths:**

1. Coherent framework unifying reasoning and generation across text, speech, and video for digital human interactions
2. Demonstrated cross-modal alignment and visual quality.

**Weaknesses:**

1. The paper's claim of real-time operation lacks substantiation, as it does not provide enough quantitative latency measurements or empirical evaluation to support this assertion.
2. Sweeping assertions such as "infinite" and "hours-long" are not backed by quantitative evidence, such as long-term consistency curves, which are necessary to validate such strong performance statements.
3. The system implementation described in the paper reads more like an engineering effort that stitches together a collection of known, established techniques and models, lacking any particularly impressive innovation.

**Questions:**

1. The so-called "global identity reference" is a common paradigm in LLM-based reference image/video generation, where the reference image is simply embedded as part of the multimodal input sequence. Therefore, how does your proclaimed approach substantively differ from these existing methods?
2. What is the actual end-to-end latency (ms) from user audio input to first video frame, and how are Thinker and Actor distributed or synchronized across GPUs?
3. Can X-Streamer work normally across different races, ages, genders and nonhuman or stylized avatars?
4. Can X-Streamer maintain coherence over 1-hour conversations without any degradation?
5. How to consider using a 2-second multimodal segment?

---

> ### Author Response · Authors · 2025-11-18
> **Author Response to Reviewer 16Le**
>
> We sincerely thank Reviewer 16Le for the insightful comments and questions. We are glad to see the reviewer's appreciation of our work as a **"coherent framework unifying reasoning and generation across text, speech, and video"** with **"demonstrated cross-modal alignment and visual quality."** Below we provide detailed responses to each of the reviewer's comments.
> ***
> **1. Real-Time Latency and Dual-GPU Inference.**
>
> We thank the reviewer for this valuable comment. We provide interactive real-time demo recordings in the supplementary material to illustrate the system’s responsiveness.
>
> The measured end-to-end latency, from receiving the user’s audio to producing the first rendered video frame, is 4.11 seconds on average. This latency primarily arises from three components:
> - Audio encoding: ~0.37 s to process the full user audio input.
> - Thinker inference: ~1.64 s to generate the first 2-second multimodal segment.
> - Actor decoding: ~1.78 s to synthesize the corresponding 2-second video clip.
>
> We note that this latency could be further reduced by adopting lower-latency LSM variants as the Thinker module, e.g., Moshi [1], a real-time spoken dialogue system with a 160ms theoretical latency.
>
> To achieve real-time throughput, the Thinker and Actor operate on two separate GPUs:
> - GPU 1, hosting Thinker, performs input audio and text encoding, and executes all Thinker inference to produce multimodal hidden states.
> - GPU 2, hosting Actor, handles autogressive video diffusion and VAE reconstruction after receiving the chunked hidden states.
>
> This parallelized design effectively minimizes bottlenecks and supports continuous streaming generation. Low-latency transmission between remote GPU servers and client devices is enabled through a WebRTC pipeline powered by LiveKit.
>
> In the revised manuscript, we will include this full quantitative latency breakdown.
>
> **2. Long-Range Consistency Evaluation.**
>
> Thank you for this constructive comment.
>
>  On our supplementary webpage, we qualitatively showcase camera-recorded 50 minute-long continuous streaming generation (the “Infinite Streamable Generation” example, presented with 16× acceleration). Visually, we observe stable and consistent streaming throughout the full duration, without noticeable degradation or error accumulation.
>
> To further quantitatively assess long-horizon consistency, we evaluate identity similarity (ArcFace score measured against the reference image) across the entire nearly 1-hour streaming output. We find that identity preservation remains consistently high over time: the average similarity is 0.7315, with 0.7845 at 1 minute, 0.7187 at 20 minutes, 0.7510 at 40 minutes, and 0.7293 at 50 minutes. These results provide clear numerical evidence that our method maintains strong long-horizon visual and identity consistency during extended real-time streaming.
>
> In the revision, we will include the full quantitative evaluation curve and provide additional examples demonstrating long-horizon interactive streaming.
>
> **3. Multimodal Segment Length.**
>
> As explained in Sec. 3.1 (“Audio-Visual Context Attentions”) and Sec. 4.1 (“Inference”), the choice of a 2-second multimodal segment is not arbitrary. It is directly determined by the intrinsic temporal granularity of GLM-4-Voice, which serves as the Thinker.
>
> GLM-4-Voice generates text and audio tokens in an empirically validated interleaving pattern: 13 text tokens followed by 26 speech tokens, which together correspond to roughly a 2-second semantic and prosodic window. This structure ensures that speech tokens are generated under the full guidance of the corresponding text tokens, yielding synchronized multimodal reasoning with optimized latency.
>
> Building upon this fixed temporal structure, we introduce the video modality by appending 384 video latents after each text–audio block, corresponding to 1.92 seconds of video. This design ensures that video generation is also conditioned on the complete text/audio context, while remaining divisible by 8 to match the 8× temporal compression of the 3D video VAE.

---

> ### Author Response · Authors · 2025-11-18
> **Author Response to Reviewer 16Le**
>
> **4. Generalization.**
>
> As demonstrated across the visual results in the main paper, appendix, and supplementary webpage, X-Streamer exhibits strong generalization across a wide range of appearances, including different genders, races, and even stylized avatars (e.g., the cartoon character in Fig. 1). It also handles varied portrait scales (face-only, chest-level portraits) and diverse environments (e.g., studio settings with microphones) and accessories (e.g., sunglasses, headphones).
>
> This generalization ability arises from several design choices: (1) our use of a VAE tokenizer, which preserves spatial details more faithfully than global 1D tokenizers (e.g., Body of Her [2]); (2) the global identity reference mechanism that stabilizes identity across streaming windows; and (3) training on a large and diverse corpus of talking videos covering a wide variety of human identities. Together, these components enable X-Streamer to generalize robustly across identities, styles, and conditions.
>
> In the revision, we will also explicitly acknowledge the current limitations: the model does not generalize well to non-human figures or full-body images, as these lie well outside the distribution of real-human portrait data used for training.
>
> **5. Novelty.**
>
> We respectfully disagree with the characterization that the work lacks novelty. The contribution of X-Streamer is not in introducing yet another backbone model, but in the architectural and algorithmic advances required to make long-horizon, real-time multimodal streaming interactions across text, audio, and video actually work, a capability that current multimodal LLMs and video generation systems fundamentally do not support.
>
> Below we highlight the three key technical contributions of our work:
>
> First, we propose a unified generative paradigm for multimodal, multi-turn interactions via a two-level Thinker–Actor architecture. This design explicitly decouples long-horizon semantic reasoning from low-level visual synthesis. The Thinker operates fully in token space to plan multimodal semantics (content, prosody, emotion), while the Actor realizes these plans in spatiotemporal space through diffusion decoding. This separation enables long-range omni-modal consistency, which cannot be achieved by monolithic one-pass models or standard LLM-conditioned pipelines. Our formulation also serves as a pioneering step toward general human world modeling.
>
> Second, at the Actor level, we enable unbounded, real-time, and stable video streaming with tight cross-modality synchronization, which to our knowledge is unprecedented in prior work. We design an autoregressive diffusion transformer operating in compressed video-VAE latent space, reinforced by diffusion forcing and a global identity anchor to maintain identity and stability over long horizons. Moreover, we carefully determine the multimodal segment duration and introduce a chunk-wise cross-attention mechanism to achieve accurate audio–lip synchronization while preserving long-range temporal coherence. These components have been extensively ablated and are all critical to the system's performance.
>
> Third, we demonstrate a fully functional real-time interactive system capable of long-horizon audio–visual streaming and interaction. Delivering this system requires substantial algorithmic and engineering innovations, including considerations around latency, consistency, synchronization, and visual fidelity. Importantly, our framework provides a cost-efficient training paradigm for extending pretrained language–speech models to the video domain: it requires only talking-head videos, without the need for expensive omni-modal triplets, yet still achieves strong cross-modal alignment in real-time settings.
>
> Demonstrating a system with these capabilities is nontrivial and constitutes a substantive research contribution, particularly in a field where streaming generation, digital humans, and world-model agents are increasingly evaluated by their real-world applicability.
>
> **6. Global Identity Reference.**
>
> We do not claim the global identity reference as a novel mechanism. Rather, we integrate it into our pipeline in conjunction with the sliding-window visual context, and we extensively ablate its necessity for long-duration consistency (Figure 5, Table 2). In our design, the global identity token is persistently conditioned across all diffusion windows. This provides a stable and continuous identity anchor even when earlier visual tokens are truncated by the sliding window. As a result, the model maintains strong long-horizon identity consistency throughout streaming generation.
> ***
> [1] Alexandre Défossez, Laurent Mazaré, Manu Orsini, Am´elie Royer, Patrick Pérez, Herv´e Jégou, Edouard Grave, Neil Zeghidour. Moshi: a speech-text foundation model for real-time dialogue. arXiv preprint arXiv:2410.00037.
>
> [2] Tenglong Ao. "Body of her: A preliminary study on end-to-end humanoid agent." arXiv preprint arXiv:2408.02879.

---

> ### Author Response · Authors · 2025-11-24
> **Follow Up on Our Response**
>
> Dear Reviewer 16Le,
>
> We are writing to kindly follow up on our rebuttal submitted last week. We would like to confirm whether our clarifications have addressed your main concerns, and whether there are any remaining points that may benefit from further explanation or additional supporting evidence. If there is anything more we can provide to assist your evaluation, please do not hesitate to let us know. Thank you very much again for your thoughtful review and your time.
>
> Best regards,
>
> Authors

---

### Author Response · Authors · 2025-12-01
**Summary**

Across the initial reviews, all reviewers acknowledged the overall value of our work as *"a coherent framework unifying real-time reasoning and generation across text, speech, and video"*. During the discussion period, reviewer J6da explicitly noted that the concerns regarding **training/inference details** and **long-horizon evaluation** were **substantially resolved** after our clarifications and additional quantitative evidence.

**1. Training & Inference.**

Reviewers 16Le, J6da, and EKMn requested clearer descriptions of the training and inference procedures. In the rebuttal, we provided detailed explanations of the pipeline, ablated alternative training strategies, and clarified component-level behaviors in streaming mode.

**2. Long-Horizon Evaluation.**

To address the requests from 16Le and J6da about “hours-long” consistency, we added new experiments, including ~1-hour continuous generation with identity-consistency curves. These results directly demonstrate the system’s stability over long horizons and satisfactorily answer the reviewers’ concerns.

**3. Scope of the Method.**

We clarified that the method is intentionally scoped to portrait-scale, real-time multimodal streaming agents. We will revise the wording to avoid any possible overclaiming and to explicitly position the work as a first modeling step toward broader human-world modeling, addressing the scope-related concerns raised in discussion.

**4. Novelty and Relation to Prior Work.**

We emphasized that the core contribution lies not in integrating existing components, but in formulating and addressing a **new scientific problem**: unified, real-time, long-horizon autoregressive text–audio–video generation. This problem is not solved by existing multimodal LLMs or video generation systems. Our Thinker–Actor architecture introduces a two-level separation between long-term semantic planning and short-term high-fidelity visual synthesis, **an architectural capability absent from prior systems**. The remaining critique from reviewer J6da reflects an assumption about prior systems that differs from the problem we formally define; we clarified this distinction and will make it even more explicit in the related-work section.

Overall, the discussion shows clear convergence among reviewers: all major technical concerns—including training/inference clarity, long-horizon stability, and scope—were substantially resolved, and the remaining critique originates from a mismatch in problem formulation. With the new quantitative evidence, methodological clarifications, and revised wording incorporated, we believe the paper offers a solid and well-scoped scientific contribution.

---

### Meta-Review · Area_Chair_84sQ · 2026-01-05

**Summary:**

This paper proposes X-Streamer. However (1) it is lack of clear scientific problem formulation and limited technical novelty (Reviewers J6da and 16Le). X-Streamer combines existing components—GLM-4-Voice as a frozen “Thinker,” diffusion forcing, chunk-wise autoregressive video generation which constructs a system-level digital human rather than a well-defined research problem; (2) this paper contains insufficient experiments (Reviewers J6da and 16Le, EKMn), including unsubstantiated assertions about “real-time” operation and “infinite” streaming, human world modeling, experiments on long-duration videos, more cases, ablation studies, in-depth analyses, and some baselines.

**Reviewer Concerns:**

Authors emphasize the novelty lays in a unified, real-time, long-horizon autoregressive text–audio–video generation. However, this paper does not present insufficient experiments about this, the “unified” is just integrating two models, and “world modeling” is not mentioned.  Authors argue insufficient experiments by implementing limited quantitative and qualitative experiments mainly on real-time, long-duration, but lack of more systematic experiments. Therefore, the concerns about novelty and experiments are not significantly addressed.

**Reviewer Scores:**

The reviewers will not raise the scores.

---

### Decision · Program_Chairs · 2026-01-26

Reject